# AR/VR Teaching-Learning Experiences in Higher Education Institutions (HEI): A Systematic Literature Review

Belen Bermejo [1], Carlos Juiz [1,*], David Cortes [1], Jeroen Oskam [2], Teemu Moilanen [3], Jouko Loijas [3], Praneschen Govender [2], Jennifer Hussey [4], Alexander Lennart Schmidt [2], Ralf Burbach [4], Daniel King [4], Colin O'Connor [4] and Davin Dunlea [4]

1 Computer Science Department, University of the Balearic Islands, 07122 Palma, Spain
2 Research Centre School, Hotelschool The Hague, 2587 AH The Hague, The Netherlands
3 LAB8 Service Experience Laboratory, Haaga-Helia University of Applied Sciences, 00520 Helsinki, Finland
4 School of Tourism and Hospitality Management, Technological University Dublin, D07 EWV4 Dublin, Ireland
* Correspondence: cjuiz@uib.es

**Abstract:** During the last few years, learning techniques have changed, both in basic education and in higher education. This change has been accompanied by new technologies such as Augmented Reality (AR) and Virtual Reality (AR). The combination of these technologies in education has allowed a greater immersion, positively affecting the learning and teaching processes. In addition, since the COVID-19 pandemic, this trend has been growing due to the diversity of the different fields of application of these technologies, such as heterogeneity in their combination and their different experiences. It is necessary to review the state of the art to determine the effectiveness of the application of these technologies in the field of university higher education. In the present paper, this aim is achieved by performing a systematic literature review from 2012 to 2022. A total of 129 papers were analyzed. Studies in our review concluded that the application of AR/VR improves learning immersion, especially in hospitality, medicine, and science studies. However, there are also negative effects of using these technologies, such as visual exhaustion and mental fatigue.

**Keywords:** AR/VR; systematic literature review; higher education; teaching-learning process

## 1. Introduction

In the last few years, technological development has influenced our lifestyle. The main purpose of technology is typically to increase productivity in industries, ease life, or improve education [1]. Integrating technology into education allows one to facilitate learning methods and improve learning performance by creating and managing appropriate technological materials. In addition, it promotes students' capacity to learn how to use new technologies in their future life [1].

In this work, we address virtual reality (VR) and augmented reality (AR) technologies used for improvements in the teaching-learning process. AR can be defined as an interactive experience in the real-world environment where computer-generated information and elements are linked to the real world [1]. The computer-generated information is virtual content that may be synthesized with the help of multiple sensors and haptic devices. In contrast to AR, VR takes place within an artificial environment and a participant becomes a part of this artificial world as an immersive and non-immersive member. Users can interact and manipulate computer-generated objects in a virtual environment with the help of gadgets such as haptic devices. It is important to note that in this work we will use the term "teaching-learning experiences" to describe the interaction between teachers and students using AR/VR technology. The term includes the experiences of teachers when teaching and the experiences of students when learning.

Some years ago, public research institutions started to finance projects related to AR/VR technologies in the teaching-learning space [1]. This is the case with the THETA

Project (KA220-HED-8C845691). The overall aim of THETA is to prepare students and professionals for a changing profession by offering a versatile, virtual context for real-life case studies, using AR/VR-enabled learning spaces.

As a result of the THETA project, in this work, we review the experiences of teaching and learning using AR/VR in Higher Education Institutions (HEI).

This work is organized as follows: related and previous works are described in Section 2. Then, in Section 3, we describe the systematic review methodology followed. The proposed classification of the selected papers and their descriptions are described in Sections 4 and 5. In Section 6, we discuss the main findings of this study, finishing with the conclusions and future work in Section 7.

## 2. Related Works

Since the integration of AR/VR technologies in the teaching-learning process, several researchers have reviewed the usability and drawbacks of using these technologies. In [2], the authors performed a systematic review of AR/VR, analyzing 88 articles from five perspectives: methods of integrating AR or VR tools into language learning, the main users of AR and VR technologies, the major research findings, why AR and VR tools are effective in promoting language learning and the future implications. As we can observe, this review was focused on language education, specifically on the used tools, and the type of education institution was not considered.

The XR (Extended Reality) challenges, opportunities, and future trends that will impact HEI were studied in [3]. The authors studied 103 millennial and post-millennial students to comprehend their facility in learning with XR technologies. In this case, the teaching-learning experiences were not considered.

Other previous reviews [4] covered the current state of universities regarding ICT integration, focusing on technologies that have been emerging in recent times, such as AR/VR. Moreover, in [5], the authors presented a systematic review of 42 papers to understand, compare, and reflect on a recent attempt to integrate immersive technologies in education, using seven dimensions: application field, the technology used, educational role, interaction techniques, evaluation methods, and challenges. In this publication, the teaching-learning experiences were not considered or evaluated. In addition, AR/VR technologies were studied as part of a set of different technologies.

In addition, [6] reports findings from a systematic mapping review from 2010 to 2020, after evaluating the use of AR applications in Science, Technology, Engineering and Mathematics (STEM) subjects. The authors did not collect information regarding teaching-learning experiences.

In [7], the authors reviewed applications intended to complement curriculum materials for K-12. They reviewed 87 research papers to perform an analysis of the design aspects of the K-12 curriculum. In [8], a systematic review of the research literature was conducted on the use of AR in e-learning contexts, with a focus on the key benefits and challenges related to its adoption and implementation. In these two research works, the teaching-learning process was not considered.

From the bibliometric point of view, in [9], the authors examined the overall research trends and productivity in the fields of VR in HEI. The main gap in this work is a bibliometrically-focused orientation which does not consider the research contents.

To the best of the authors' knowledge, this paper is the first attempt to review the experiences of teaching-learning using AR/VR technologies in HEIs.

## 3. Systematic Literature Review: Methodology

As we stated previously, this work aims to review the experiences of teaching and learning using AR/VR in HEIs. To achieve the proposed aim, a Systematic Literature Review was performed [10]. For this, the methodology proposed by [11] was followed. The first step was to search in selected scientific databases, comprising Google Scholar, Science Direct, and IEEE Xplore, in this case. After this, we applied a selection process to the

obtained sample. To finish, we classified the obtained information considering bibliometric indicators [12].

### 3.1. Search Process

The first step was searching current research works in the scientific databases. Google Scholar, Science Direct, and IEEE Xplore were selected as the most suitable for our field of study. The identified keywords for this search were: XR, VR, AR, teach, learn, HEI, and university. Considering that we attempted to review the experiences in the teaching and learning process, we built the following logical expression: (("XR" OR "VR" OR "AR") AND ("teach*" OR "learn*") AND ("HEI" OR "university" OR "universities")). Moreover, we extended this logical expression using: (("mixed reality" OR "virtual reality" OR "augmented reality") AND ("teach*" OR "learn*") AND ("high education institute" OR "university" OR "universities")). Executing a search with these logical expressions from 2012 to 2023 in all the databases, we obtained a set of 1328 initial results.

### 3.2. Selection Process

The second step was the selection of aligned research works from the initial pool. We removed all the duplicate works found in the different databases. Then, we selected the works coming from indexed journals, obtaining a set of 636 results.

To filter the pool of papers published in indexed journals, we manually filtered for relevance by reading the abstracts. This filter sought to determine if the paper contributed to the experiences in the teaching-learning process using AR/VR. If not, the paper was discarded. It is important to note that we did not consider the application of AR/VR in specific fields. We only considered the teaching-learning experiences. After this filtering, we obtained a final set of 129 papers, which were analyzed in this work.

### 3.3. Bibliometric Information

The selected 129 papers were analyzed for their bibliometric data, obtaining information such as the publication year (see Table 1). It can be observed that from 2012 to 2017 the number of papers describing teaching-learning experiences using AR/VR represented a limited percentage of those selected. Nevertheless, from 2018 to 2022, the number of AR/VR experiences increased. This fact probably reflected the maturity and/or the improvement of AR/VR technology, in terms of its devices and software. The most papers describing AR/VR experiences were found in 2021. Moreover, more than 62% of the identified works were published between 2020 and 2022.

**Table 1.** Analyzed papers classified by year.

| Year | Number of Works | Percentage (100%) |
| --- | --- | --- |
| 2012 | 3 | 2.26 |
| 2013 | 0 | 0.00 |
| 2014 | 2 | 1.50 |
| 2015 | 2 | 1.50 |
| 2016 | 7 | 5.26 |
| 2017 | 5 | 3.76 |
| 2018 | 13 | 11.28 |
| 2019 | 14 | 10.53 |
| 2020 | 16 | 12.03 |
| 2021 | 37 | 27.82 |
| 2022 | 28 | 21.05 |
| 2023 | 4 | 3.01 |

The country where the publication originated from is also considered in the literature analysis. In Table 2, the classification of the selected works by country is shown. We can observe that China, Australia, the UK, Germany, and Spain are the countries with the

highest publication count regarding AR/VR teaching-learning experiences, with Spain being the country with the most publications on the subject.

**Table 2.** Analyzed papers classified by country or region of publication.

| Country | Reference | Country | Reference |
|---|---|---|---|
| China | [1,13,14] | Spain | [15–20] |
| Iran | [3] | USA | [13,21,22] |
| Serbia | [3,23] | Finland | [24,25] |
| Romania | [3,23] | Nigeria | [26] |
| Saudi Arabia | [3,27] | Thailand | [28] |
| Australia | [6,7,29,30] | Korea | [31] |
| Kazakhstan | [8] | Norway | [32] |
| Egypt | [11] | Japan | [33,34] |
| South American countries | [35] | Zambia | [36] |
| Jordanian | [37–39] | Indonesia | [40] |
| UK | [41–44] | Middle East | [45] |
| Mexico | [24] | Taiwan | [46] |
| France | [42] | | |
| Germany | [42,47,48] | | |

## 4. Proposed Classification

In this section, the classification of the research works is shown. Since this work aimed to study teaching-learning experiences using AR/VR in HEIs, the following features were considered: the corresponding academic discipline, the educational level (bachelor or master), the experience for teachers and students, and the method for data collection.

### 4.1. Classification by Discipline

Teaching-learning experiences using AR/VR are present in all the knowledge fields, from more technical ones to healthcare-based ones. Table 3 shows the classification of the analyzed research works by discipline.

**Table 3.** Analyzed papers classified by applied discipline.

| Discipline | Reference | Discipline | Reference |
|---|---|---|---|
| Language learning | [2,49] | Dental | [21,27,47,50] |
| Teaching in general | [3,11,16,23,32,37,42,44,51–66] | Chemistry | [13,67–69] |
| Medicine | [5,15,19,22,30,33,35,36,70–80] | Political theory | [81] |
| STEM | [6,82,83] | Computational thinking | [70,84] |
| English foreign language | [8,85–87] | Technology | [25] |
| Geography | [88] | Distance education | [89] |
| Computer hardware | [90] | Forensic | [39] |
| Engineering | [29,91–93] | Mathematics | [94–96] |
| Pharmacy | [14,41] | Computer games | [43,97–101] |
| Architectural education | [38,102] | Psychology | [103,104] |
| Physics | [105] | Digital production studies | [106] |
| Applied science | [24] | | |

We can observe that medicine and general education are the fields with the highest publication count regarding AR/VR. In most of the works from medicine, the authors propose techniques involving holograms and 3D objects to learn anatomy, interventions with patients, and surgical techniques.

In this context, "general education" refers to various knowledge fields in terms of improving any teaching-learning experience from the student's point of view. The studies in this category evaluate the feelings and emotional experiences of students using AR/VR for learning, especially in social and language learning.

### 4.2. Works Classified by Educational Level

Considering the educational level, we considered the bachelor's and master's curricula at HEIs. As Table 4 shows, the papers are classified by educational level. We can observe that the AR/VR teaching-learning experiences are used more in bachelor's programs than in masters. This difference can be attributed to the number of students enrolled in these programs and to the nature of the programs. To analyze the teaching-learning experiences, a large number of students is required. For this, bachelor's studies may be more suitable than master's ones.

**Table 4.** Analyzed papers classified by education degree.

| Bachelor | Master |
|---|---|
| [11,14,15,19,20,22,25,26,29,33,57,61,68,70,72,73,79,88,89,93,102,104,105,107–111] | [11,52,61,89] |

### 4.3. From the Experience of Teachers or Students

As stated in previous sections, this paper aimed to analyze the teaching-learning experiences using AR/VR. The main stakeholders involved in these experiences are teachers and students.

We classified the literature considering these stakeholders (Table 5). Some studies considered the teaching-learning experiences from the student's perspective, with a focus on student learning. Other studies focused on the teachers' perspective and improvement of the teaching process, or reported on both perspectives, when the attempt is to improve the whole experience, from students' and teachers' point of view. Table 5 shows that most of the works are related to the student's experience, with a focus on use of AR/VR to improve the student learning process.

**Table 5.** Analyzed papers classified by experience perspective.

| Students | Teachers | Both |
|---|---|---|
| [2,4,5,8,11,13–20,22–24,27,29,33–37,39–52,58–69,71–79,82–91,93–101,103,104,106–128] | [53,92,129,130] | [28,54–56,79] |

### 4.4. From Methods for Data Collection

This work aimed to explore the experiences using AR/VR in HEIs. To analyze the teaching-learning experience, a set of techniques to collect data is necessary, such as questionnaires, case studies, systematic reviews, surveys, interviews, and experimental studies.

- A questionnaire is a list of questions or items used to gather data from respondents about their attitudes, experiences, or opinions.
- A case study is a research approach that is used to generate an in-depth, multi-faceted understanding of a complex issue in its real-life context.
- A systematic review is a scholarly synthesis of the evidence on a presented topic using critical methods to identify, define and assess research on the topic.
- A survey is defined as the act of examining a process or questioning a selected sample of individuals to obtain data about a service, product, or process.
- An interview is a structured conversation where one participant asks questions, and the other provides the answers.
- Experimental studies are those where researchers introduce an intervention and study its effects.

As Table 6 shows, questionnaires and surveys are the most used techniques to evaluate these experiences because, through a questionnaire and a survey, data can be collected from each stakeholder. A considerable number of works also used experimental studies. In the same manner, data can be collected from the observation of an experiment which incorporates AR/VR experience.

**Table 6.** Analyzed papers classified by data collection method.

| Survey | [3,8,9,11,13,16–18,21,22,29,32,34,35,37,41,45,48,50,52,55,56,60, 64,66,68,76,77,83,85,86,108–111,113,114,120,121,123,124] |
| --- | --- |
| Systematic Review | [4,91] |
| Questionnaire | [19,20,24,27,30,38,39,46,47,66,67,69,71,82,84,85,87,90,92,93,95, 97,100–102,104,119,122,128,130] |
| Interview | [26,33,36,44,70,88,96,99,103,125,129] |
| Experimental- Study | [14–16,25,40,42,43,49,51,57,59,61–63,73– 75,78,80,81,94,98,112,115–118,126,127] |
| Case Study | [53,65,72,79] |

## 5. Main Research Work Description

In this section, we will explain the main features of the considered main research work from the teacher's and students' points of view.

### 5.1. From a Student's Experience Point of View

The main reason for involving AR/VR technologies in the teaching-learning process is to improve the quality of learning and engage students. In this section, we will describe the main research works regarding the students' experiences.

In [11], the authors proposed procedures to run an educational XR lab safely and contribute to the conversation about how to carry out research involving users in XR under pandemic restrictions. This paper discussed the experiences of using XR for immersive learning research in higher education at a Norwegian university, running an XR lab during a national lockdown. The authors conducted a short survey of master's and bachelor's students in July 2020 with 14 students. They also reported qualitative observations and reflections from being XR student supervisors and teachers during the pandemic. The feedback from students using the lab and their projects show that they received adequate support to complete their research despite the complex circumstances. The lesson for XR lab operators is that they will need to have the ability to pack, hand out, and track equipment at very short notice under new circumstances such as the pandemic. XR equipment is portable and powerful, so quality work can be carried out outside the XR lab, if necessary.

Regarding English learning, [8] investigated the effects of using open social VR for university English as a Foreign Language (EFL) studies. Data from multiple sources were collected and analyzed (surveys, video-recordings and oral reports). Findings reveal that the participants perceived the social and physical presence afforded by the VR environment positively. They enjoyed the interactions with international interlocutors via VR technologies in the digital context. In the same manner, in [85], the authors investigated the effects of using a 3D VR reality game on university EFL students' development of vocabulary and cultural knowledge. In this case, 25 students participated in the VR game-based language learning, while a control group of 24 students followed the regular curriculum of the university EFL course. The feedback was collected using a questionnaire and an online survey of students' perceptions and attitudes. As a result, the study suggested that VR game-based language learning is a new, effective way to promote students' vocabulary and cultural knowledge. Moreover, in [89], the authors performed a study that sought to determine the perceptions of bilingual and ESL pre-service teachers using VR in their studies and to provide an understanding of their VR mobile applications' experiences. A content analysis was performed with 27 students to analyze their reflections, and three categories were determined.

In [88], the authors discussed the technological capabilities of VR in education as a highly developed form of computer modeling. To measure the impact of VR technology on the quality of e-learning, the authors experimented during an online physical geography course. The experiment involved 60 third-year students (Abai Kazakh National Pedagogical University, Immanuel Kant Baltic Federal University, and Aldar University College). The results showed a high degree of adaptability of VR technology in education as well as an increase in the respondents' requirements for the quality of subsequent academic training.

Thus, a significant impact of immersive technology's evolution on the demands on e-learning quality can be noted.

In [90], the authors developed a VR program about introductory computer hardware and invited freshmen from two universities in China to take part in the experiment. The behavior of 132 students was recorded through a questionnaire. From the obtained results, the relationship between the students' behavior and their assigned VR study group was found, demonstrating that the VR program is effective at attracting the students' curiosity and increasing the understanding of computer hardware in the UCF course. Similarly, in [35], the authors described the development and evaluation of a pilot set of VR anatomy resources at the University of Dundee. Students were exposed to a collection of 3D anatomical models in VR to evaluate their usefulness and consider the potential for adoption of this technology for anatomy education. In this study, 18 participants evaluated the pilot through a survey. Moreover, in [51], the authors performed a study to explore the application of VR technology in welding courses, developing materials to teach welding practice and implementing experimental teaching to identify their effectiveness. Qualitative and quantitative research and analysis were performed with 34 first-year students of the electric welding practice course. The results show that the introduction of virtual-reality-assisted teaching of welding-related courses in various universities of science and technology was positive. Another important work in the engineering field is [91], where the authors reviewed evidence from past research that VR is an excellent tool in engineering education. From the performed review, the authors deduced that VR has positive cognitive and pedagogical benefits in engineering education, which ultimately improve the students' understanding of the subjects, performance and grades, and education experience.

In the area of civil engineering, in [29] the authors presented a study on the application and assessment of using virtual reality for infrastructure management education in civil engineering. A group of 69 senior-year undergraduate students was evaluated through a structured rubrics-based assessment. The results showed that subjects demonstrated a higher capacity for concentration in VR environments and found the VR experience to be easy to use.

In [37], the authors explored innovations in music curricula in colleges and universities in the era of digital multimedia VR technology. The survey on the current situation of the implementation of digital multimedia VR teaching in colleges provided a more detailed basis for the reform of the music curriculum. It also showed how digital multimedia VR teaching tends to improve the implementation of music in colleges. In addition, [41] describes the use of mobile-based AR technology to develop an interactive learning module about contraceptive devices and medicines and the measurement of its acceptability and usability by undergraduate pharmacy students. Students answered a survey to collect information about the usability and acceptability of AR for learning. The results showed that the majority of students reported that AR is a useful resource for learning about medicine compared to more traditional methods. Moreover, in [38], the authors suggested a virtual environment technology as a tool to develop a new educational approach. They developed an application to deal with building construction using VR. The study sample was selected from the population of building construction students at Jordan University of Science and Technology (JUST). A structured questionnaire was designed and distributed to the students. The results showed that with the use of VR software students can achieve learning objectives better than with more traditional teaching methods.

In the same manner, in [105], the authors presented a study that aims to evaluate the students' experience using VR tools to support their learning about three-dimensional vectors in an introductory physics university course. The authors developed an experimental research design (control and experimental group). The results showed that students evaluated the VR tool as having a positive impact on their course content learning and as a valuable tool to enhance their learning experience. In addition, in [24], the authors described first-year healthcare students' expectations regarding teaching, studying, and learning in such an environment. A questionnaire was distributed among 47 students

from two different universities of applied sciences in Finland. From the analyzed data, the authors observed that students have quite high expectations of activities that take place in VR and SBLW. Adult learners in particular seem to have high expectations compared to younger students.

The transmission of scientific knowledge is studied in [42], where authors determined the difference between VR images and traditional audiovisuals in terms of their usefulness to transmit scientific knowledge. The authors performed a quasi-experiment with 302 participants from public, private, urban, and suburban universities of Chihuahua (Mexico). The results obtained confirmed that attributes for playful and experimental first-person learning, due to the combination of 3D and 360º video, were effective. In another study, the authors described the first randomized controlled multi-center trial to assess VR to teach infection prevention and control measures [15]. A set of 110 participants of third-year medical students from Grenoble Alpes University, Imperial College London, and the University of Heidelberg were randomized into two groups (intervention and control). In [21], the authors explored the usability of a VR application in training the practical skills of dental students. Using the System Usability Scale (SUS), the authors verified the validity of the VR application in this study. A subsequent survey was delivered to collect the participants' perception and evaluation of the VR system application in training the practical skills of the prospective dentist. Overall, the use of VR in dental education achieved the expected outcomes. Most students identified VR training as an enjoyable learning process, and it could be repetitively experienced without further costs. In the same manner, in [27], the authors assessed the usability of VR technology in dental education at Kermanshah University of Medical Science (Iran). In this case, 50 six-year dental students were assessed using a test. Additionally, a questionnaire was used to assess the usability of VR technology and students' satisfaction with it. All faculty members confirmed the usability of VR technology in dental education. The majority of students (76%) were highly satisfied with the use of this technology in their learning process.

In [13], VR lab experiences for organic chemistry were developed at NC State University as an accessibility tool for students who were unable to attend in-person laboratories due to disabilities. A survey was distributed among 141 students. The students who completed VR laboratories reported more positive affective experiences than they anticipated, including little frustration or confusion in the laboratory. In [16], the authors explained the benefits of VR in transforming learning and student experiences in classrooms. The experiment involved students using an educational VR experience that features an immersive narrative that puts students in the center of a historical moment in World War Two. The authors used a survey, an online test, and focus group questions. The results showed that there is a potential for immersive narrative VR experience to provide students with new experiences and provide both cognitive and affective benefits.

The objectives of [52] were to investigate knowledge about AR technology, the design and production of digital objects in AR, and to assess the usefulness of this technology in higher education. A survey was distributed to 186 participants (students) and four discussion groups were held at the University of Sevilla and the University of Barcelona. The results showed the innovative, interesting, and playful nature of AR as a resource in the teaching and learning processes, where students develop a series of skills and become "prosumers" rather than consumers of their digital objects. The paper also highlighted the importance of technological and pedagogical training for future professionals. Moreover, in [81], the main purpose of this paper was to combine VR technology with ideological and political theory courses in colleges and universities. The data were collected from experiments. After a month, the students were tested in the subject. It was found that VR classroom teaching can stimulate students' interest in learning, promote the understanding of knowledge, as well as support the establishment of emotional attitudes and values.

The assessment of learning was performed in [70], aiming to determine if AR/VR is as effective as tablet-based applications and whether these models allowed enhanced student learning, engagement, and performance. During the assessment, 59 participants

were randomly allocated to one of the three learning models (VR, AR, or tablet-based) and completed a lesson on skull anatomy. The students' perception of each learning mode and any adverse effects experienced were recorded, showing that there were no significant differences between assessment scores in VR, AR, and tablet-based learning.

In [102], the authors examined the benefits and potential applications of integrating AR technology into landscape design education to create a more rewarding educational environment. This provides an interesting learning atmosphere and develops students' knowledge of the landscape design process. An experiment was conducted on the fourth-year architecture students at Port Said University. Then, a questionnaire was formulated and distributed to the students to examine their feedback. The results showed that the integration of AR with traditional teaching methods was perceived as being useful and having a positive impact on landscape design education. Similarly, in [25] the authors evaluated the effect of holding traditional training courses and VR-based training courses on sustainable behavior. It was a quasi-experimental study, which included 105 students from Iranian universities. The results showed significant differences in the mean score of sustainable behaviors in the post-test phase between the two experimental groups and the control group, indicating that VR can be an effective way of learning sustainable behavior.

Regarding gaming, in [26], the authors focused on designing and implementing a VR game-based application to support computational thinking. This study followed the design science research methodology to design, implement, and evaluate the prototype of the VR application. An initial evaluation of the prototype was conducted with 47 computer science students from a Nigerian university. As a result, this study made a significant contribution to positioning computational thinking in the HEI context and provided empirical evidence regarding the use of educational VR mini-games to support students' learning achievements.

The effects of using an experimental methodology with the students of a new Master of Library Science (MLS) course were studied in [89]. A set of 362 students from the Library Information Science program were evaluated using a pre- and post-test. The results showed that students who received a VR orientation expressed more optimistic views about the technology. The majority of students also indicated a willingness to use VR technology for learning for long periods. In [130], the authors investigated the impact and design of a multiuser, VR-supported teaching simulation, in comparison with a live classroom teaching simulation. They used a mixed method including qualitative data collection. The data analysis showed that VR-based simulation can supplement and work as an alternative to the live classroom simulation to support participatory teaching development.

### 5.2. From the Teacher's Experience Point of View

University teachers and professionals are the main stakeholders involved in the teaching-learning process using AR/VR technologies. Therefore, it is important to analyze their experiences of using AR/VR to teach and/or to engage learners.

In [92], the authors analyzed the assessment of engineering professors of different nationalities and universities regarding the use of VR technologies in the classroom. A questionnaire was designed and distributed among 279 university professors from different engineering schools in South American countries. Engineering teachers provided high evaluations of VR as a didactic tool, but they also showed a certain lack of knowledge and specific training regarding its use. In [53], the authors examined pre-service teachers' perceptions of VR and their beliefs about its capacity to be used as a teaching and learning tool. A case study was conducted at an urban university in Australia, which involved a total of 41 university professionals. The authors observed that the perception of the participants regarding VR in their teaching was positive, in terms of its potential to engage learners, the immersive potential of the platform, and the scope of VR to offer students experiences they might otherwise not have through other learning tools.

In [129], a study to evaluate teachers' perceptions of the use of AR for heritage teaching was developed. The objectives of this study were to identify teachers' existing knowledge

about AR, to evaluate educational strategies that teachers value most in AR apps for teaching and necessary AR functions, to determine desired technical and functional characteristics, and to identify any significant differences between the two groups. For this, the authors developed a questionnaire distributed among 347 teachers. The authors thus sought to contribute to the increasing implementation of AR apps in heritage education, which promotes the understanding, enjoyment, experience, and knowledge of heritage.

A final research work which considered the teaching experience is [130], which identified the faculty awareness of the economic and environmental benefits of AR for sustainability in Saudi Arabian universities. The authors distributed a questionnaire among 228 academic and e-learning department staff from these universities. From the obtained results, the authors concluded that for academic and e-learning department staff, the use of AR in higher education has positional environmental and economic sustainability benefits.

*5.3. From Student's and Teachers' Experience Points of View*

It is important to highlight that the teaching-learning process has involved two stakeholders simultaneously: teachers and students. For this, some works were developed considering the AR/VR experience of both.

In [28], the authors proposed a mobile AR (MAR) system aimed to support students in the use of milling and lathe machines at a university manufacturing laboratory. The authors distributed a survey among 16 students and teachers from the mentioned laboratory. Based on the obtained results, the authors stated that students, laboratory technicians, and teachers had a positive opinion and good acceptance of the use of the MAR system in the manufacturing laboratory.

Additionally, in [54], the authors proposed an approach to support the introduction of VR in education delivery. They conducted a pilot study that focused on embedding VR as a medium to teach empathy within higher education milieus. The study began by conducting a pilot faculty development workshop to provide an understanding of VR and ways it can be embedded as a pedagogical approach to support CV design. In this study, five members from a local university were recruited to participate. The pilot study suggested that embedding VR into the courses is a feasible approach that provides an engaging learning environment that is effective for teaching an array of personal skills.

Moreover, in [55], the authors decided to systematically document their approach to the selection, development, and implementation of four new VR teaching applications. They developed an intrinsic case study, outlining the process and critical elements that shaped the selection of suitable teaching content, software development, hardware solutions, and implementation. Pilot courses were developed by the teaching staff at Newcastle University, and tried by their students, and data was collected using pre- and post-surveys.

In [56] the authors applied a mixed-methods approach to gain insight into end-user acceptability, value areas, barriers, and opportunities for the adoption of XR teaching at an Australian university. A university-wide online survey and targeted interview sessions with XR technology users showed a general readiness for the broad adoption of XR technologies in university education. Whilst existing teaching applications were described as "successful", relatively few applications were sustainably integrated into the curriculum. The collected data highlighted the existing barriers to the successful transition from cases of individual use of XR tools to broader adoption across university institutions.

## 6. Discussion

In this section, we will discuss the most noteworthy results of the presented review.

It is plausible that the COVID-19 pandemic has accelerated the use of AR/VR technologies. However, this impact is temporally coincident with further advances in new AR/VR technologies. This acceleration resulted in more than 62% of the studied research works being published after 2020. That is, after 2020, AR/VR technologies were increasingly included in the learning-teaching process. However, we cannot distinguish the causes by reading the contents of the papers included in this SLR, with the exception of [11,30],

where the analysis is centered on the satisfaction of students and lessons learnt during the pandemic. As stated in the introduction, our SLR research work aims to review the experiences of teaching and learning using AR/VR in HEIs, to study the extension of applicability of these technologies in the last 10 years, and also to know the disciplines in which there are more mature experiences.

Health and medical faculties are the most interested in the research and use of the AR/VR teaching-learning process. In these areas, it is useful to use holograms and 3D objects rather than physical bodies. The experiences from medical fields can be extrapolated to other fields, such as ICT (hardware courses) and language learning. General education is another focus of interest in the AR/VR teaching-learning process. As we know, AR/VR technologies are one of the latest innovations in the teaching-learning process. For this, it is necessary to evaluate the pros and cons of using them in any field.

Regarding the educational levels in HEIs, bachelor's programs feature more in AR/VR published research on teaching-learning initiatives than master's and doctoral programs. This may be related to the number of students, which is generally higher in bachelor's programs than in master's and doctoral courses, as well as the general nature of the subjects. Therefore, it is more convenient to evaluate the teaching-learning experience in bachelor's programs. Along the same lines, students' experiences are more often evaluated than the teachers' expectations. That is, students are considered the most important stakeholder in AR/VR teaching-learning experiences research to date.

Finally, considering the collected data, questionnaires, surveys, and experimental studies are the most used data collection methods. They allow researchers to formulate ad hoc questions and tend to be easier in terms of the collection and analysis of data.

## 7. Conclusions

In this paper, we have summarized the latest published research works regarding the experiences of teaching-learning experiences using AR/VR in HEI. In addition, we have reviewed the previous works to find the gaps in the past research. Moreover, we reviewed the main concepts regarding AR/VR technologies and the THETA project.

The main contribution of this work is the categorization of the related literature considering the applied discipline, the education degree, the country, the experience for teachers and students, and methods for data collection. From the different classifications, we conclude that research on AR/VR technologies is most often applied to medical studies for bachelor's degrees, and the collected data are typically analyzed using surveys and questionnaires. Moreover, this paper could help other researchers to evaluate and create teaching-learning experiences using AR/VR technologies in any knowledge field. Meanwhile, the paper has identified the more mature experiences in HEIs and the disciplines where these technologies were applied.

As a future avenue for research, this paper has established a gap in terms of opportunities regarding teaching-learning experiences. All the evaluated experiences have indicated the starting point of using AR/VR in many fields of knowledge. Data collection methods based on questionnaires and surveys can be used in other studies. Moreover, the type of hardware used in the teaching-learning experiences could be considered in future works. Another possible future research study should be to identify factors that may increase the applicability of AR/VR in HEIs in the future and the reasons to adopt these new approaches.

**Author Contributions:** Conceptualization, B.B., C.J. and D.C.; methodology, B.B.; validation, C.J., D.C., J.O., T.M., J.L., P.G., J.H., A.L.S., R.B., D.K., C.O. and D.D.; formal analysis, B.B.; investigation, B.B.; resources, B.B. and C.J.; data curation, B.B.; writing—original draft preparation, B.B. and C.J.; writing—review and editing, C.J., J.O., T.M., P.G., J.H. and A.L.S.; supervision, C.J.; project administration, C.J., J.O., T.M. and R.B.; funding acquisition, C.J. All authors have read and agreed to the published version of the manuscript.

**Funding:** This research was funded by (Call 2020 Round 1 KA2-Cooperation for innovation and the exchange of good practices) European Commission, grant number KA203-5EAF909E.

**Institutional Review Board Statement:** Not applicable.

**Informed Consent Statement:** Not applicable.

**Data Availability Statement:** Not applicable.

**Conflicts of Interest:** The authors declare no conflict of interest.

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
