# Peer review of "AR/VR Teaching-Learning Experiences in Higher Education Institutions (HEI): A Systematic Literature Review"

_informatics, doi:10.3390/informatics10020045_

Round 1

Reviewer 1 Report

This paper has potential but needs improvements. This study barely defines augmented reality (AR) and does not define virtual reality (VR), even though that is necessary since those definitions have changed in the literature of the past few decades and likely will change in the future. In fact, a key element missing in this meta-study is specifically listing what hardware was used in each study. Lumping studies using Google Cardboard to studies using an Oculus Quest Pro into one category of VR doesn't make sense.

Also, the authors need to define what they mean by "teaching-learning experience" with examples early on, especially if that is supposed to be the original component they are adding in this study.

The list of papers seems a bit limited and it is odd (and not explored or explained at all) why Spain has the highest number of papers. It seems a lot of papers might have been missed if the researchers didn't search for "higher education" as a keyword search term. I rarely see "HEI" used in publications in the USA. This seems evident in the table below where only two references are from the USA. Additionally, searching for "virtual reality" instead of just "VR" likely leads to different results. 

The tables in Section 3 are formatted so poorly they are hard to follow and sometimes don't make sense. For example, why are Physics, Chemistry, Applied Science, and Engineering not part of STEM in Table 3? Table 4 needs to be reformatted. There is no gap between the Bachelor and Master numbers.

Section 5 is a long list briefly describing the studies but it isn't well organized and very little is done by the authors to "connect the dots" between the studies.

Section 6 had the potential to be interesting, but it makes some baseless claims and doesn't go into any detail on the analyses of these papers to add much to our understanding of VR in higher education.

The argument that COVID-19 led to more VR papers due to the timing is not supported. In fact, as the authors note elsewhere, the pandemic made it more difficult for students to access this hardware. The rise in VR publications likely stems from the new VR technology that became available. For example, the untethered Oculus Quest came out in 2019. The authors should, at minimum, describe other reasons why the surge in VR/AR publications occurred instead of merely assuming a causal relationship between COVID-19 and VR research since they both happened in 2020.

This discussion of what higher education fields use VR and why they do it is one of the few original aspects of this research paper. The authors really should spend much more time on this rather than just glossing over it quickly.

Similarly, the discussion of the number of participants in each study is curious and interesting, yet they authors say very little here. How many studies actually have a statistically valid number of respondents? Are most VR papers not that valid? This would be interesting to know!

I would strongly recommend that the authors double-check for papers using full-length keywords like "higher education" and "virtual reality" to be sure they aren't missing papers. I'd also suggest organizing Section 4 so that it has papers sorted more logically. Finally, and most importantly, I'd spend time writing substantially more in Section 5 to explore the interesting things they found in looking at the list of papers. Finally, the abstract must be rewritten to allude to these interesting findings.

Author Response

Dear reviewer,

Thank you in advance for providing us with constructive feedback. We explain all the modifications we did in the paper in the attached file.

Best regards,

All authors.

Reviewer 2 Report

The paper represents a valuable overview of literature in AR/VR for teaching and learning using appropriate methods and offering clear recommendations for research. Lines 332-334 require translation to English.

Author Response

(The authors gave the same response as above.)

Reviewer 3 Report

The paper presented a systematic review of the use of AR/VR in education. It has analysed 129 papers and presented the findings and classifications of the previous work based on year, country, discipline, degree ...etc

While the paper presented a very nice classifications and findings, I believe that it can be improved with the following suggestions:

1. The teacher/students' experience on VR/AR will mainly depends on the technology used, however, there is no classification of the used technologies and the impact of the AR/VR technology on the users' experience.

2. I expected that, beside the limitation identified for the reviewed studies to see suggestions for further research in the field, but I cannot find it or I missed it 

3. User experience can be impacted by many factors such as usability, reliability of the technology, perhaps a definition of experience or experience factors tested by each reviewed paper would be very helpful 

4. The authors did not discuss the limitations of their studies.

Above are some suggestions to improve the work, although I believe that it is a very interesting work

Author Response

(The authors gave the same response as above.)

Reviewer 4 Report

This paper describes a systematic review of virtual reality (VR) and augmented reality (AR) as information technologies to improve the teaching-learning process in universities and other higher education centers. The review was made from 2012 to 2022.

The paper is very well written and organized. The review is very thorough and well described, following a methodology based on 3 steps: search scientific databases, selected the samples and classify the results considering bibliometric indicators.

Some suggestions that may improve the text:

- Be careful with the hyphenation of words throughout the text.

- References appear in the text without any order. References must be placed in ascending order.

- Define the acronym ARLEs (line 86).

- Add a blank line between line 186 and 187.

- Add a comma after the reference in the sentence: "In [12] 47 computer ..." (line 295).

- Remove the blank line between lines 317 and 318.

- A paragraph written in Spanish between lines 332 and 334. Change to English.

Author Response

(The authors gave the same response as above.)

Round 2

Reviewer 1 Report

This is a somewhat improved version of the original manuscript. The authors are claiming to merely be listing the various studies and not attempting to analyze them (e.g., the statistical validity, etc.). There is still an unfounded claim that is not supported. The authors claim this change was made in Section 3.3, but I don't see it there. I'll reiterate my concern below. I am willing to look past the other inadequacies, but this claim that the pandemic led to more VR studies is unsupported by the data and probably false. You can't just say two things happened at the same time and therefore one caused the other!

"The argument that COVID-19 led to more VR papers due to the timing is not supported. In fact, as the authors note elsewhere, the pandemic made it more difficult for students to access this hardware. The rise in VR publications likely stems from the new VR technology that became available. For example, the untethered Oculus Quest came out in 2019. The authors should, at minimum, describe other reasons why the surge in VR/AR publications occurred instead of merely assuming a causal relationship between COVID-19 and VR research since they both happened in 2020."

Author Response

Thanks a lot for your second review and comments, we modify some issues based on your suggestions. More detailed responses in the next paragraphs:

This is a somewhat improved version of the original manuscript. The authors are claiming to merely be listing the various studies and not attempting to analyze them (e.g., the statistical validity, etc.).

R: the aim of this study as we stated in the introduction is to review the experiences of teaching and learning using AR/VR in Higher Education Institutions (HEI) to know more about to prepare in the next future students and professionals for a changing profession by offering a versatile, virtual context for real-life case studies, using AR/VR-enabled learning spaces. The analysis is based on reading what others have done in a systematic literature review and classifying their experiences from the students’ and teachers’ viewpoints.

There is still an unfounded claim that is not supported. The authors claim this change was made in Section 3.3, but I don't see it there. I'll reiterate my concern below. I am willing to look past the other inadequacies, but this claim that the pandemic led to more VR studies is unsupported by the data and probably false. You can't just say two things happened at the same time and therefore one caused the other!

R: you are right is not in the 3.3 is now in the discussion section. Is an interesting issue to discuss and needs further investigation in future work when more experiences have been collected (more than 3 years) and see what are the factors, if there are more and their relationship, … but this is far from an SLR study to perform in next years to know this. We have modified it accordingly in the discussion and not in the conclusions.

"The argument that COVID-19 led to more VR papers due to the timing is not supported. In fact, as the authors note elsewhere, the pandemic made it more difficult for students to access this hardware. The rise in VR publications likely stems from the new VR technology that became available. For example, the untethered Oculus Quest came out in 2019. The authors should, at minimum, describe other reasons why the surge in VR/AR publications occurred instead of merely assuming a causal relationship between COVID-19 and VR research since they both happened in 2020."

R: you are right we modify the discussion section in this way.